# Unpleasant but effective: Newspaper coverage of cancer screening and cancer in the Netherlands from 2010 to 2022

Martin-Pieter Jansen[1]*, Inge Stortenbeker[1], Hanneke Hendriks[2], Suzan Verberne[3], Gert-Jan de Bruijn[4], Enny Das[1]

1 Centre for Language Studies, Radboud University, Nijmegen, The Netherlands, 2 Behavioural Science Institute, Radboud University, Nijmegen, The Netherlands, 3 Leiden Institute of Advanced Computer Science, Leiden University, Leiden, The Netherlands, 4 Department of Communication Studies, University of Antwerp, Antwerpen, Belgium

☺ These authors contributed equally to this work.
* martin-pieter.jansen@ru.nl

## Abstract

Participation rates in cancer screening programs (CSPs) have shown a declining trend, and research suggests that news media reports may contribute to public opinion. Therefore, we aimed to understand how Dutch news media report on CSPs. We mainly focused on breast, colorectal, and cervical CSPs but did not exclude reports on other cancer types. Through a systematic content analysis 5,503 news articles from 2010 to 2022 were analyzed for key characteristics such as topic and reported cancer type. Results showed that most news reports framed CSPs as effective and beneficial for public health. The reliability of screening methods was sometimes criticized. In these cases, reports discussed overdiagnosis or medicalization. Although reports were positive about CSPs' effectiveness, they were sometimes negative about organizational, psychological, and physiological aspects. Early detection and diagnosis of cancer are portrayed as having benefits that outweigh the costs. These findings show that news media often describe CSPs as a 'necessary evil' and that participation may be inconvenient and stressful, but that early detection and diagnosis of cancer are benefits that seem to outweigh this necessary evil.

## Introduction

The World Health Organization (WHO) distinguishes three main cancer screening programs (CSPs) as having the most potential to be effective in early detection of (pre-, or early stage) cancer: breast, cervical, and colon [1]. These programs represent important detection tools in the early or precursor stage of cancer, and participation is thus recommended [2,3]. However, while participation rates of at least 70% are needed for these programs to be effective at a population level [4,5], Dutch participation rates between cancer types differ and are relatively low: 46.1% for cervical,

**Data availability statement:** Raw files, study protocol, analysis plan, and codebook can be found on OSF: https://osf.io/z7y6u.

**Funding:** Disclosure statement: This research is part of the SENTENCES (Social mEdia aNalysis To promotE cancer Screening) project and is funded by ZonMw under grant project number 555004205. Project leader for this project is prof. dr. HHJ Das, who is also an author of this paper. https://projecten.zonmw.nl/nl/project/sentences-social-media-analysis-promote-cancer-screening.

**Competing interests:** The authors have declared that no competing interests exist.

70.7% for breast, and 68.4% for colorectal cancer screening [6–9]. In addition, overall participation seems to show a declining trend [10]; therefore, it is vital to understand the potential factors contributing to participation rates.

In the Netherlands, the National Institute for Public Health and the Environment (RIVM) is responsible for CSPs. It is essential that the RIVM effectively informs the public about the considerations of taking part in CSPs. Nevertheless, not all health information comes from flyers and pamphlets. Another way that people can inform their health-related decision-making are news media reports [11]. Research shows that frames used in news stories can encourage readers to better their health behaviors [12]. Moreover, news coverage of health topics has been shown to influence the public's health-related beliefs, attitudes, and behaviors, making news reports a relevant means for health related decision-making [13].

Given the relevance of media reports for (predictors of) health-related behaviors, it is vital to understand how media reports on cancer screening programs. To the best of our knowledge, no research has analyzed the content of Dutch news media reports regarding cancer screening systematically. Therefore, this work aims to answer the question of how the Dutch news media reported on CSPs. While the main emphasis of this study is on the largest CSPs in the country (breast, colorectal, and cervical), we do not exclude reports on other cancer types, as we aim to compare the relative volume of news about cancer in general and CSPs. In doing so, this work adds to the current literature by comparing news reports on cancer and cancer screening, and the three CSPs in the Netherlands.

## Theory

As news reports can contribute to public opinion and societal debates [14], we investigate news media reports on CSPs through a content analysis that focuses on different aspects of written news reports.

## Volume, characteristics, and content

Knowledge is one of the determinants of partaking in CSPs, and news media reports can be a source of information contributing to such health-related knowledge [10,15]. Earlier work shows that news media reports on CSPs have the potential to complement physician advice to partake in screening programs [16], and also increase awareness and knowledge [17]. Accordingly, research shows that if a topic occurs in news reports more often, this could potentially influence public perceptions of the occurrence of the topic [18,19]. In other words, news media might cultivate the issue, making it appear more salient in the perceptions of receivers [20,21]. Moreover, topics that are shared in news articles often overlap with topics that are discussed on social media platforms, which shows the relevance of investigating news articles in order to map the broader societal discourse about CSPs [22,23]. In the Netherlands, 29% of people over the age of 13 read newspapers [24], and there is a positive relation between peoples age and their newspaper reading [25]. This partially overlaps with the fact that people that are invited to participate in CSPs are usually of higher age [26,27].

However, it is not only occurrence in the news that shapes the public's perceptions and understanding of topics. The way news media reports frame topics can have significant implications for how the public perceives the risks and benefits associated with these topics [28]. A key aspect of framing in news media is that the emphasis in articles can be placed on certain aspects of an issue while other issues could be omitted. Hereby, news media can selectively highlight parts of problems.

Earlier work shows that news coverage of cancer screening highlights the need to protect and expand access to screening, but at the same time might undermine confidence in established and proven screening programs [29]. Moreover, in the case of colon cancer screening, loss framing (i.e., a frame that highlights the undesirable consequences that can be associated with noncompliance [30,31]) in screening promotion has been found to lead to greater message-evoked fear and lower intentions to take part in screening [32]. Accordingly, other research shows that gain-framed messages seem to be more effective compared to loss-framed messages when it comes to promoting cancer prevention behavior [33]. In contrast, a study focusing on CSPs in Japan shows that cancer screening messages are found to be more persuasive when they are framed in terms of the costs of not taking part in screening (i.e., loss frames) but only a small number of loss frames actually appear in Japanese newsletters [34]. It should be noted that these contrasting results could be moderated by cultural differences, as other research shows that white Europeans are more persuaded by gain-framed health messages, while East-Asian participants have a stronger prevention focus and are more strongly persuaded by a loss-framed message [35]. Moreover, the framing of topics in particular ways can help to shape beliefs and attitudes on health issues [14,36], and cancer-related knowledge [37,38]. Thus, although existing research shows different results regarding different types of framing, it is important to recognize the role that frames in reports have in relation to cancer screening. One way of describing frames in a quantitative way is to investigate the pros and cons that are linked to the topic of discussion [39,40]. Therefore, we aim to investigate how cancer screening is discussed on the content level by looking at the pros and cons that are described in relation to screening practices.

Taking the above into account, the first aim of this work is to describe cancer related news, based on its volume and characteristics. Moreover, we aim to describe the frames used if pros and cons of screening are presented, and subsequently, which arguments are used for these presentations. Combined, this is formulated in our first research question:

RQ1: What are the volume, characteristics and content of newspaper reports about cancer screening?

## Differences across time and cancer types

In the Netherlands, the most prevalent types of cancer are cancer in the digestive system and skin, followed by breast and lung cancer and finally, cancer in the male genital organs. General tendencies of cancer type incidence prevalence remains stable in the period of this work [41]. However, prevalence of a specific type of cancer does not predict news portrayals [42]. A bigger factor that determines if and how news media report on topics is how 'newsworthy' a topic is. Research shows that in the case of Malaysian cancer news coverage, monthly distributions are relatively constant, calling cancer news an essential part of routine health news production [43].

Research shows that news coverage over time differs as well, potentially contributing to overrepresentations, for instance regarding breast cancer. While breast cancer is one of the most common types of cancer, research shows that it is overrepresented compared to its relative frequency and mortality rates [41,42]. Work on illness representations demonstrates that people develop their own (mis-)representations of the illness, which may partially be based on overrepresentations in news media [44]. This shows that indeed, the public perceptions of cancer do not always represent the reality of the disease. Which is in line with the cultivation of news media that is described earlier in this work [20,21]. However, to the best of our knowledge, no work that investigates news media reporting on cancer in the Dutch media landscape has yet been published. Therefore, the second aim of this work is to describe the coverage of cancer screening in written news media from 2010 till 2022. The research question for this aim differs from the one we preregistered, as we do not investigate the extent, we focus on that in RQ3. The third aim is to investigate differences in reporting between the three different types of cancer screening. These aims lead to the second and third research questions:

RQ2: How does the coverage in newspaper articles change over time?

RQ3: To what extent and how does the coverage in newspaper articles differ between different types of cancer screening?

### Contextualization

Besides the different frames that are discussed earlier (RQ1), news media also use different types of sources in articles to contextualize stories that are reported. Earlier research argues that newspapers and journalists are able to make use of contextual frames in the neutral genre of news reporting to propose specific interpretations of the facts [45,46]. Earlier work shows that in the United States, national medical institutions and medical schools at research universities are often cited in news related to cancer [47]. However, the same study indicates that, in terms of story types, the majority of cancer news coverage consists of personal profiles of individuals with cancer, followed by the previously described reports on cancer research [47]. Other work shows that different frames could be used with different sources. For instance, frames of the causality of cancer risk factors mostly are accompanied by medical institutions, healthcare providers and scientists as the predominant sources in those articles [48]. Moreover, other research shows that news media reporting on prostate cancer screening in Australia framed this CSP as desirable and minimized discussing the adverse effects [49]. Other work shows that in line with this, American newspaper articles on screening in general are supportive of screening and do not discuss the factors that should be considered regarding screening [29]. However, these authors argue, that the newsworthiness of new methods could also backfire, as coverage of new screening technologies could undermine public confidence in the current and established CSPs, that are proven to be effective [29].

The above shows that it is not only important to unravel which frames regarding cancer screening are discussed in news media reports, but also to investigate how these frames are contextualized to better understand how cancer screening is portrayed in news media reports. We will investigate information about CSPs in news articles based on which sources are used in the article (e.g., scientists or people with experiential knowledge) and how these sources discuss the CSPs. Further, we will assess whether frames are positive or negative about current (vs. innovative) measures via the presented pros and cons in the articles. Therefore, the fourth and final aim of this work is to investigate how information about cancer screening is contextualized in written news articles, leading to the fourth research question:

RQ4: How is cancer screening portrayed in news reports with respect to the sources cited and the frames employed?

## Method

This preregistered content analysis systematically assessed Dutch news reports about cancer screening. The preregistration, raw data files, study protocol, analysis plan, and codebook can be found on the Open Science Framework (https://osf.io/z7y6u, https://doi.org/10.17605/OSF.IO/Z7Y6U). We received ethical approval by the Ethics Assessment Committee Humanities of Radboud University under approval number 2022–3900 on November 11th, 2022. As no human participants took part in this study, informed consent was not applicable. We hand-coded the characteristics, risk and effectiveness frames, and the pros and cons of screening. The study protocol including the codebook was submitted to OSF prior to data analysis.

### Search procedure

We used the news database Lexis Nexis to download full-text articles on cancer screening published in local, regional, and national newspapers from January 2010 to October 2022. The search string included synonyms for the various cancer types, combined with strings related to screening and/or specific methods included in the national screening program (e.g., mammography, swab) (see SI1). However, to compare the relative volume of news about cancer to that about cancer screening, we also registered the total number of news articles that included specific cancer strings without words

 

related to screening to obtain an estimate of the volume of news about cancer in general. See SI1 for an overview of the specific news media outlets and the full search string. We downloaded 12,236 articles in total, of which 976 were removed because they pertained to non-Dutch newspapers. We excluded English-written news articles.

## Duplicate detection

In the Netherlands, there are media companies that publish multiple newspapers. For example, DPG Media Group owns national, regional, and local ones. Because local and regional newspapers can publish identical articles in various outlets, we used the Rouge metric to detect duplicate files [50]. Rouge is an evaluation metric that computes similarity statistics between two texts, either using a unigram (single word similarity: Rouge-1) or bigram (similarity between two co-occurring words: Rouge-2) and the longest common subsequence (sentence-to-sentence similarity: Rouge-L). We tested which of the three measures with varying thresholds for similarity scores performed best at detecting duplicates (i.e., least chances of failing to detect duplicates or incorrectly marking single files as duplicates) in a smaller test set containing the first 1,500 characters of the full text of a testing set of 100 randomly selected newspaper articles. The Rouge-L metric with a threshold of.85 similarity appeared to perform best at detecting duplicates, that is, the least chances of failing to detect duplicates or incorrectly marking single files as duplicates (see SI2 for an overview of the performance scores). In total, 5,757 duplicates were detected and removed.

## Coding procedure

The final dataset included 5,503 articles, of which 924 were manually coded using systematic sampling (i.e., every 5th article, sorted by date of publication). Fig 1 shows the selection process.

   A codebook describing the relevant coding criteria was developed using an iterative process. The criteria were developed deductively based on previous research [51–53] and by consulting experts from the field (i.e., parties involved in cancer screening, such as RIVM and KWF), and inductively by carefully reading a random subsample of potentially relevant articles. The main researcher (IS) developed the first version of the codebook, after which the first coder (LS) finalized the coding categories. When the research team agreed to the final version of the codebook, a second coder (NB) joined the team. During the training of the second coder, some minor final adjustments were made to the codebook (e.g., coders were instructed to note all the pros and cons of screening rather than categorizing each of them, which made the coding easier and thus more reliable) [54]. Inter-coder reliability was calculated using Cohen's Kappa ($\kappa$) [55]. The double-coding procedure followed a stepwise approach; the coders first coded a subset of 30 training articles to check inter-coder reliability. The reliability of some of the categories appeared only fair ($\kappa > .25$), which resulted in extra training for the coders. The second set of 30 articles provided substantial Kappa values ($\kappa > .60$), after which the remainder of the subsample ($n = 78$) was double-coded. The training round of 30 articles is not included in the kappa calculation. They were carefully checked and served as a 'best practice' database for the coders to support them in the coding process. The subset is included in the final data analysis. After the definite Kappa calculation (see kappa's below), the first coder coded the entire subset.

## Coding operationalization and reliability

A Python script (see OSF) was written to automatically retrieve the date of publication, title, name of the newspaper, and number of words for each article. The coders first indicated whether an article was relevant for inclusion ($\kappa = .91$). Articles from Dutch newspaper articles were considered for inclusion when they reported on cancer screening programs currently in the Dutch populational screening program (namely, cervical, breast and colon cancer). We also included articles lung and prostate cancer screening as they were (and still are) under debate to be included in the populational screening program. Current methods included current screening practices (i.e., pap smear, mammogram, stool test) relevant for cervical, breast, and colon cancer. Debated methods included innovations such as curved paddles for mammograms.

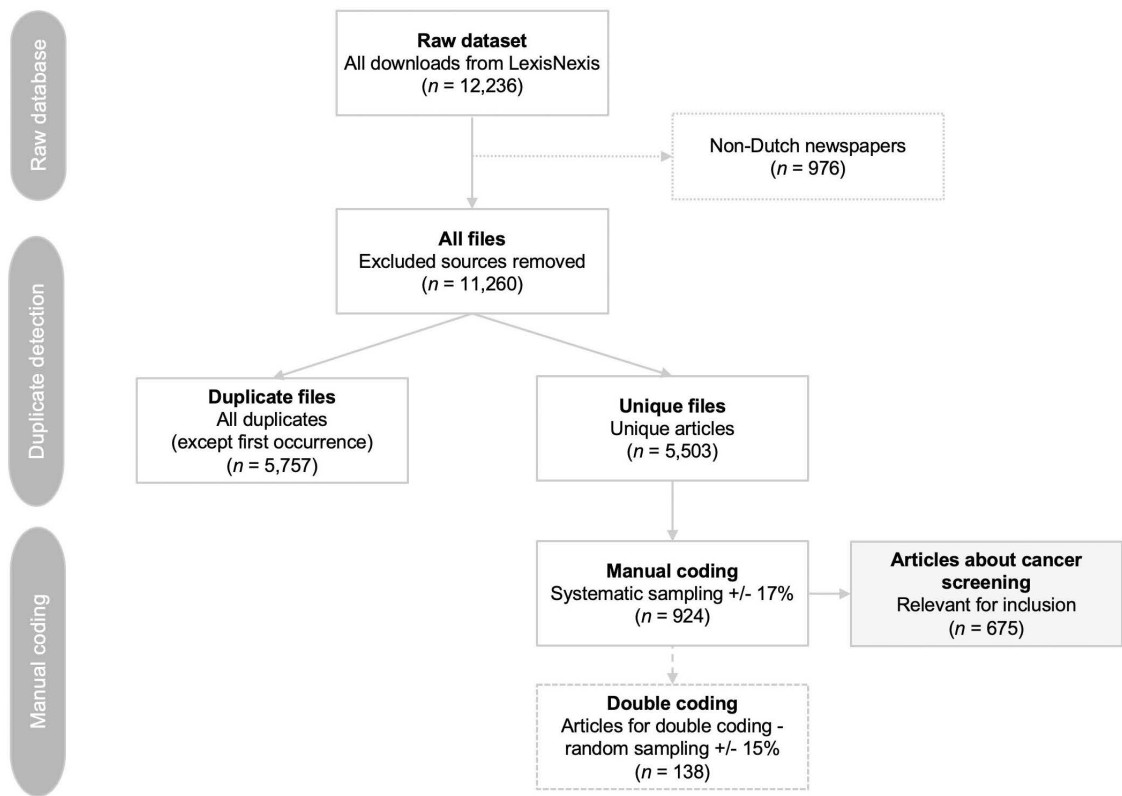

**Fig 1. Flow chart of the news articles selection procedure.**

Next, the **characteristics** of the news reports were coded. *News genre* denoted whether the article had the form of a news item, column, letter, interview, background article, or other genre (κ = .78). *News topic* referred to broad news categories, i.e., health(care); politics; science; technology and economy; culture, media, sports, and entertainment; COVID-19; and other (κ = .82). It was coded whether cancer (screening) was the *main or side issue* of the news article (κ = .69), whether the *main topic* was related to cancer, cancer screening, or early detection (κ = .77), and what was the *dominant type of cancer* reported in the article, i.e., cervical, breast, colon, prostate, or lung cancer, or cancer screening in general (κ = .96). Whereas the coders had to choose one option for the previous coding categories, two categories allowed the selection of more than one subcategory: *preventive measures* included vaccination, lifestyle (e.g., healthy eating, condom use), alternative screening (e.g., full body scan), and surgery (e.g., mastectomy) (κ = 76.), and *sources* that were mentioned or cited were categorized as individuals with experiential knowledge, a healthcare providers, research, politics, health organizations, or other sources (κ = .71).

Finally, the content of news about cancer consisted of two main categories, i.e., news frames and the pros and cons of screening, that each were divided into various subcategories. News *framing* consisted of three subcategories: the extent to which cancer screening was framed as **effective** (κ = .78), the framed **risks** of getting cancer (κ = .78), and the framing of potential **consequences** of cancer (κ = .77). Table 1 lists the news framing categories with examples from the corpus.

The main coder (LS) extracted all **pros and cons** about current screening methods and possible innovations mentioned in the articles (i.e., each argument was coded as a *pro* or *con*, and whether it was related to *current* screening practices or *innovations*). The main researcher (IS) coded all pros and cons post hoc, distinguishing argument types related

**Table 1. Framing categories as defined in the codebook.**

| Frame | Subcategory | Operationalization | Example[a] |
|---|---|---|---|
| **Screening effectiveness** | *Explicitly effective* | Screening is explicitly framed as effective in reducing the risks of cancer. | "Fewer diagnoses due to screening", "The swab saved my life" |
| | *Implicitly effective* | The underlying assumption is that screening is effective, without explicit arguments supporting this claim. | "Thermography cannot replace mammography" (i.e., mammography would be effective in detecting breast cancer) |
| | *Effectiveness disputed* | The article includes any sentence or reference that (even slightly) questions the effectiveness of screening – even though the main conclusion is that screening is effective. | "Women with very dense breast tissue are more prone to developing tumors. These are harder to detect on a normal mammography" |
| | *Neutral* | No explicit effectiveness frames. | "50-year-olds are eligible for screening" |
| **Cancer risk** | *High* | The risk of getting cancer is framed as high. | "Breast cancer is the most common cancer type in women" |
| | *Low* | The risk of getting cancer is framed as low. | "Only 800 women per year are diagnosed with cervical cancer" |
| | *Neutral* | No statement about cancer risks, or neutral frames without contextualization of risk. | "800 women per year are diagnosed with cervical cancer" |
| **Cancer consequences** | *Severe* | The consequences of cancer are framed as high | "Breast cancer is death cause number one" |
| | *Not severe* | The consequences of cancer are framed as low | "I see so many ex-patients walking around healthily" |
| | *Neutral* | No statement about cancer consequences, or neutral frames without contextualization of the consequences | E.g., prevalence without consequences: "Approximately 18.000 women per year receive the diagnosis of breast cancer" |

[a]Examples derived from the corpus.

to screening criteria, screening organization, physical impact of screening, and psychological context. Explanations and examples are presented in Table 2.

## Statistical approach

Descriptive statistics (*n* and %) provide insights into the volume, characteristics, and content of news media (RQ1). Time-series plots were used to assess how media coverage may change over time (RQ2). We plotted absolute frequencies and rolling averages (the mean value of one month/quarter before, after, and at the point of measurement) and detected key news events that corresponded with peaks in news coverage. Chi-square statistics were used to assess the contextualization of information in cancer screening news (RQ3), and possible differences in news media coverage for cervical, breast, and colon cancer, excluding prostate and lung cancer (RQ4). Standardized residuals (<−1.96 or > 1.96) were used to assess which cells differed significantly. Python was used for duplicate detection [50] and file management. R [56] was used for data preparation, the dplyr [57] and tidyr [58] packages were used for analyses, and ggplot2 [59] was used for visualization. The analysis files can be found on OSF.

## Results

### Volume, characteristics, and content of news (RQ1)

The *volume* of news about CSPs and cancer per cancer type from 2010 to October 2022 is displayed in Fig 2. It can be observed that news reports about breast cancer and breast cancer screening were at least twice as frequent compared to the other cancer types, and while news about cervical cancer was less common, a relatively larger proportion was about screening (i.e., 23.1%). Regarding news *characteristics*, most reports were published in regional newspapers. The main genre was news items, and most of the news content was related to health and health care; more characteristics can be

**Table 2. Argument types of pros and cons.**

| | Pros | Cons |
|---|---|---|
| **Screening criteria**<br>*The main criteria for assessing the effectiveness of cancer screening* | Screening allows early detection, better treatment options, is cost-effective | Screening leads to overdiagnosis, overtreatment, medicalization, is unnecessary |
| **Screening organization**<br>*Practical issues surrounding the organization of screening for individuals and/or the national screening program organization* | Screening is cheap, easy to organize, NL has the capacity to do so, easy to plan a screening appointment | Screening is expensive, lack of capacity (either or not due to COVID-19), is not feasible, difficult to plan a screening appointment |
| **Physical impact**<br>*The physical impact an individual could experience from undergoing cancer screening* | Screening is relatively pleasant or has only a low burden on people | Screening hurts, involves radiation exposure, is burdensome or unpleasant, could lead to complications |
| **Psychological context**<br>*The psychological context to individuals undergoing screening or not* | Screening provides relief, certainty, people are motivated | Screening induces fear, taboo, leads to uncertainty or an illusion of certainty, people lack motivation |

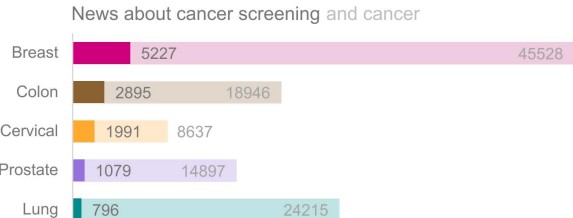

News about cancer screening and cancer

| | | |
|---|---|---|
| Breast | 5227 | 45528 |
| Colon | 2895 | 18946 |
| Cervical | 1991 | 8637 |
| Prostate | 1079 | 14897 |
| Lung | 796 | 24215 |

**Fig 2. Volume of news about cancer screening (dark bars) and cancer (light bars).**

found in Table 3. Most articles had cancer (screening) as the main topic (74.5%), of which 336 articles mainly described the current national CSP. Table 4 describes the absolute and relative frequencies of other news characteristics (i.e., cancer types, preventive measures, and sources) and frames of cancer screening.

Regarding the pros and cons of screening, a total of 1,064 pros and 1,079 cons of screening were retrieved, varying from 1 to 54 arguments per article ($M = 4.1$, $SD = 5.0$). The majority of the pros and cons were about current screening methods ($n = 1,875$, 87.5%). More than half of the arguments ($n = 1,144$, 53.4%) related to specific aspects associated with screening such as early detection, (over)diagnosis, and reliability; 611 (28.5%) related to the screening organization such as costs and capacity; 195 (9.1%) described the physical impact of screening, e.g., the (lack of) pain or radiation exposure; and 193 (9.0%) were about the psychological context of screening involving fear, taboo, relief and motivation.

To conclude, over half (58.6%) of the news reports framed cancer screening as effective. A quarter of news articles frame screening as implicitly effective, suggesting that screening would be perceived as effective without additional argumentation needed to support the claim. In approximately one in five news articles, some kind of doubt is expressed about its effectiveness. Though this proportion may seem high, only a few articles explicitly reject screening for the early detection of cancer; many simply mention a potential downside (e.g., the possible chance of a false positive or negative) but still conclude that screening benefits the public health. While the risks of getting cancer are often presented neutrally (e.g., no risk mentioned or statistics are provided without any frames), its consequences are more often mentioned (e.g.,

**Table 3. Characteristics of news articles (N = 675).**

| Characteristics | N (%) |
|---|---|
| **Newspaper reach** | |
| National | 218 (32.3) |
| Regional | 279 (41.3) |
| Local | 178 (26.4) |
| **News genre** | |
| News items | 441 (65.3) |
| Interviews/personal narratives | 146 (21.6) |
| Columns/opinion | 47 (6.9) |
| Background articles | 23 (3.4) |
| Letters | 9 (1.3) |
| Other genres | 9 (1.3) |
| **News content** | |
| Health and healthcare | 314 (46.5) |
| Policy | 93 (13.8) |
| Science | 88 (13.0) |
| Culture | 71 (10.5) |
| COVID-19 | 52 (7.7) |
| Technology | 33 (4.9) |
| Other | 24 (3.6) |

**Table 4. Frequencies of content and frames of news about cancer screening (N = 675).**

| | N (%) | | | N (%) |
|---|---|---|---|---|
| **Cancer type** | | | **Screening effectiveness** | |
| Breast | 329 (48.7) | | Explicitly effective | 221 (32.7) |
| Cervical | 82 (12.15) | | Implicitly effective | 175 (25.9) |
| Colon | 150 (22.2) | | Disputed | 126 (18.7) |
| Prostate | 28 (4.2) | | Neutral | 153 (22.7) |
| Lung | 10 (1.5) | | | |
| Cancer in general | 76 (11.3) | | | |
| **Preventive measures** | | | **Cancer risk** | |
| Alternative screening | 53 (7.9) | | High | 115 (17.0) |
| Lifestyle | 52 (7.7) | | Low | 22 (3.3) |
| Vaccination | 17 (2.5) | | Neutral | 538 (79.7) |
| Surgery | 17 (2.5) | | | |
| **Sources** | | | **Cancer consequence** | |
| Health care provider | 175 (25.9) | | Severe | 320 (47.1) |
| Science | 160 (23.7) | | Not severe | 15 (2.2) |
| Health organization | 159 (23.6) | | Neutral | 340 (50.4) |
| Politics | 69 (10.2) | | | |
| Personal | 142 (21.4) | | | |
| Other | 61 (9.0) | | | |

NB: It is possible that some subtotals are lower or higher than the total of the category because a frame could occur multiple times in an article (e.g., cancer consequence the subtotal is higher than the total frequencies and preventive measures is lower).

sickness or death). The analysis of pros and cons provides a more fine-grained picture of the type of pros and cons that are mentioned in news articles about screening and goes beyond the discussion of screening effectiveness alone. For instance, though one can argue that screening induces medicalization (con), this does not necessarily mean that the screening method is perceived as ineffective for the early detection of cancer. The total number of pros and cons appear to be equally divided, and most arguments are about screening criteria. The extent to which the type of arguments differs for pros and cons will be discussed in section "contextualization of news (RQ4)".

**Media coverage over time (RQ2)**

Fig 3 demonstrates the volume of media coverage about cancer screening from 2010 to 2022. No peaks in the news coverage can be detected when considering news articles about all cancer types (3.1), but specific news events can be related to peaks in the volume of news for the different cancer types (3.2–3.5), except prostate cancer (3.6). Many of the news events relate to the cancer screening program, e.g., the announcement and introduction of colon cancer screening, or the advice for introducing a home kit for the cervical cancer screening program. No controversial news events existed, except for speculations of conflict of interest in relation to the cervical cancer screening program that received some media-attention. SI4 provides a brief description of each news event.

**Differences between cancer types (RQ3)**

Table 5 shows the frequencies of frames and the pros and cons of screening for breast, cervical, and colon cancer (lung and prostate cancer excluded). CSPs differed significantly in how their effectiveness ($\chi^2(6) = 16.92$, $p = .010$) and risks of cancer are framed ($\chi^2(4) = 13.24$, $p = .010$). Cervical cancer was more likely to be framed as implicitly effective and having a low risk compared to chance. No other significant differences in effectiveness and risk frames were observed, and no differences were detected in how the consequences of cervical, breast, and colon cancer were framed ($\chi^2(2) = .09$, $p = .95$) ("Not severe" is removed from the chi-square analysis due to low frequencies. A Fisher's Exact test with this sublevel included produced the same non-significant result).

The type of pros and cons that were discussed in the news also differed for the different cancer types ($\chi^2(6) = 57.78$, $p < .001$). News reports about cervical cancer were relatively more often about the psychological context and organization of screening, and less about screening criteria and physical impact. Psychological context for example included women's low motivation or the taboo of undergoing a pap smear, and screening organization related to issues such as the introduction of a home kit or a conflict-of-interest case (see also SI4 for a description of the news event). Breast cancer news relatively more often concerned the physical impact instead of psychological context of screening, e.g., the pain and radiation exposure women experience during a mammogram. No significant differences in the type of arguments for colon cancer were observed.

To conclude, differences exist in news about breast, cervical and colon cancer screening. Breast cancer screening news reports frequently discuss the physical impact (e.g., pain). For cervical cancer, arguments often relate to organizational issues of screening (e.g., introduction of the home kit) and psychological aspects (e.g., taboo and lack of motivation) but screening itself is often implicitly framed as effective. For colon cancer screening, no specific patterns in news reports were observed, except that its effectiveness may be disputed more often (but please note this is only a trend, no significant difference).

**Contextualization of news (RQ4)**

The contextualization of news on cancer screening was measured using two constructs. First, we wanted to determine whether different sources framed cancer screening differently. Second, we aimed to gain insights into the context in which the pros and cons of screening were presented.

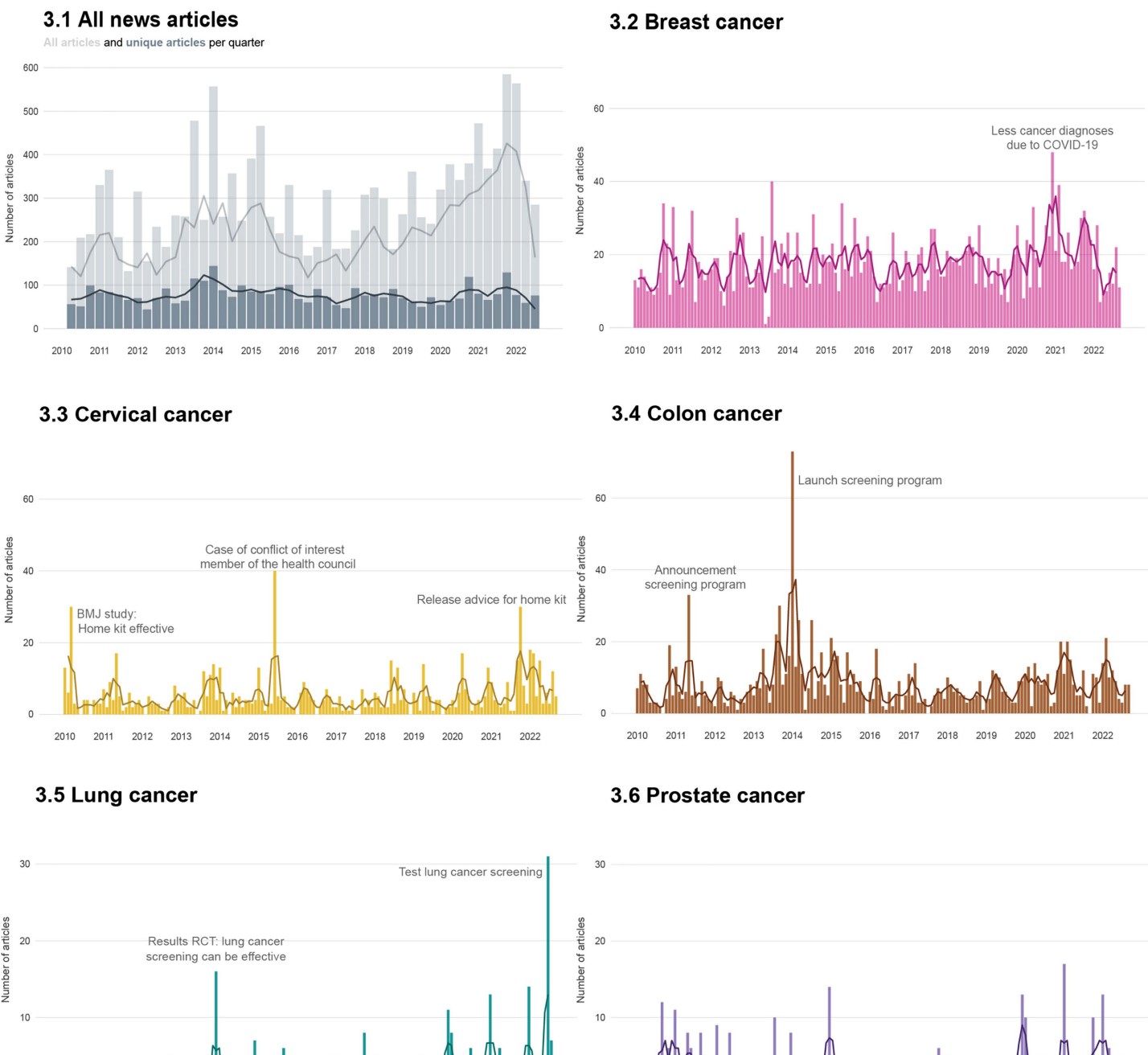

**Fig 3. Cancer screening coverage over time for all news per quarter (3.1) – differentiated between single articles and duplicates, and the current screening program consisting of breast (3.2), cervical (3.3), and colon (3.4) cancer, and early detection of lung (3.5) and prostate (3.6) cancer per month.** The absolute frequencies (bar graph) and rolling averages (line graphs) are shown.

**Table 5. Frequencies of news frames (N = 561) and pros and cons (N = 1,702) regarding breast, cervical, and colon cancer screening.**

| | Breast cancer | Cervical cancer | Colon cancer | Total |
|---|---|---|---|---|
| | n (%) | n (%) | n (%) | N (%) |
| **Screening effectiveness** | | | | |
| Explicitly effective | 110 (33.4) | 29 (35.4) | 52 (34.7) | 191 (34.0) |
| Implicitly effective | 70 (21.3) | 30 (36.6)⁻ | 37 (24.7) | 137 (24.4) |
| Disputed | 52 (15.8) | 8 (9.8) | 32 (21.3) | 92 (16.4) |
| Neutral | 97 (29.5) | 15 (18.3) | 29 (19.3) | 141 (25.1) |
| **Cancer risk** | | | | |
| High | 42 (12.7) | 11 (13.4) | 29 (19.3) | 82 (14.6) |
| Low | 6 (1.8) | 7 (8.5)⁻ | 6 (4.0) | 19 (3.4) |
| Neutral | 281 (85.4) | 64 (78.1) | 115 (76.7) | 460 (82.0) |
| **Cancer consequences** | | | | |
| Severe | 148 (45.0) | 38 (46.3) | 65 (43.3) | 251 (44.7) |
| Not severe | 6 (1.8) | 0 (0.0) | 4 (2.7) | 10 (1.8) |
| Neutral | 175 (53.2) | 44 (53.7) | 81 (54.0) | 300 (53.5) |
| **Total articles** | **329 (58.6)** | **82 (14.6)** | **150 (26.7)** | **561 (100)** |
| **Pros and cons** | | | | |
| Screening criteria | 425 (53.5) | 130 (38.4)⁻ | 315 (55.5) | 870 (51.1) |
| Screening organization | 230 (28.9) | 143 (42.2)· | 160 (28.2) | 533 (31.3) |
| Psychological context | 48 (6.0)⁻ | 47 (13.8)· | 48 (8.5) | 143 (8.4) |
| Physical impact | 92 (11.6)· | 19 (5.6)⁻ | 45 (7.9) | 156 (9.1) |
| **Total pros and cons** | **795 (46.7)** | **339 (19.9)** | **568 (33.4)** | **1702 (100)** |

NB: Superscripts indicate whether frames are significantly more (·) or less (⁻) frequent compared to chance.

## Sources

As reported in Table 6, healthcare professionals, science (scientists and/or scientific evidence), health institutes, and individuals with experiential knowledge were cited most often in news articles about cancer screening (21.4–25.7%), while politics or other sources appeared less often (9–10.2%). SI3 shows the distribution of frames among all the sources. To test for significant differences between sources, we compared how individuals with professional (healthcare professionals and science) or personal (individuals with experiential knowledge) involvement in cancer screening were framed in terms of screening effectiveness, cancer risk, and consequences. The frequency distributions per frame are presented in Table 5.

Effectiveness frames differed significantly for sources with professional and personal involvement ($\chi^2(9) = 109.17$, $p < .001$). Cancer screening was more likely to be framed as implicitly effective for personal sources, whereas news articles that cited professionals more often disputed screening effectiveness. When no personal or professional sources appeared, screening effectiveness was more likely to be framed neutral. Cancer risks were framed differently for different sources ($\chi^2(3) = 31.07$, $p < .001$), as well as the consequences of cancer ($\chi^2(3) = 93.53$, $p < .001$) (Low risk and not severe consequences excluded due to low frequencies). Risk of cancer was more likely framed as high when both personal and professional sources were used, and more likely neutral when no personal or professional source appeared. Consequences of cancer were more likely framed as severe for personal sources or both personal and professional sources, and more often neutrally when neither of the sources were cited.

In conclusion, news reports that cite individuals with personal involvement tend to frame CSPs as effective and generally describe cancer risks and consequences more often, whereas professional sources tend to have a more critical stance towards the effectiveness of cancer screening.

**Table 6. Frequencies of news frames for sources with personal or professional involvement (N = 675).**

| Type of framing | Professional n (%) | Personal n (%) | Both n (%) | None n (%) | Total N (%) |
|---|---|---|---|---|---|
| **Screening effectiveness** | | | | | |
| Explicitly effective | 68 (30.4) | 34 (32.4) | 11 (29.7) | 108 (34.9) | 221 (32.7) |
| Implicitly effective | 54 (24.1) | 46 (43.8) ↑ | 10 (27.0) | 65 (21.0) | 175 (25.9) |
| Disputed | 76 (33.9) ↑ | 13 (12.4) | 11 (29.7) | 26 (8.4) ↓ | 126 (18.7) |
| Neutral | 26 (11.6) ↓ | 12 (11.4) ↓ | 5 (13.5) | 110 (35.6) ↑ | 153 (22.7) |
| **Cancer risk** | | | | | |
| High | 43 (19.2) | 24 (22.9) | 15 (40.5) ↑ | 33 (10.7) ↓ | 115 (17.0) |
| Low | 11 (4.9) | 4 (3.8) | 4 (10.8) | 3 (9.7) | 22 (3.3) |
| Neutral | 170 (75.9) | 77 (73.3) | 18 (48.7) | 273 (88.4) | 538 (79.7) |
| **Cancer consequences** | | | | | |
| Severe | 111 (49.6) | 81 (77.1) ↑ | 28 (75.7) ↑ | 100 (32.4) ↓ | 320 (47.4) |
| Not severe | 4 (1.8) | 8 (7.6) | 2 (5.4) | 1 (0.3) | 15 (2.2) |
| Neutral | 109 (48.7) | 16 (15.2) ↓ | 7 (18.9) ↓ | 208 (67.3) ↑ | 340 (50.4) |
| **Total** | **224 (33.2)** | **105 (15.6)** | **37 (5.5)** | **309 (45.8)** | **675 (100)** |

NB: Superscripts indicate whether frames are significantly more (↑) or less frequent (↓) compared to another cell or chance. "Low" cancer risk and "not severe" cancer consequences are marked in grey since they were excluded from the chi-square analyses due to low frequencies.

## Pros and cons

We assessed whether the pros and cons differed for current (vs. innovative) measures and different argument types. A significant difference was observed for the number of pros and cons reported for the current methods (e.g., mammography for breast cancer screening) versus screening innovations (e.g., curved paddles or MRI) ($\chi$2(1) = 17.95, $p$ < .001). When screening innovations were discussed, arguments were more often in favor of screening ($n$ = 166, 15.6%) than against it ($n$ = 102, 9.5%). For the current methods, the pros and cons were presented equally.

As shown in Fig 4, we found a difference in the argumentation used for the pros and cons ($\chi$2(3) = 318.79, $p$ < .001). Screening criteria were more likely to be presented positively ($n$ = 771, 72.5%), whereas the organizational aspects of screening ($n$ = 408, 37.8%), psychological context ($n$ = 150, 13.9%), and physical impact ($n$ = 148, 13.7%) were more likely to be negative. This shows that although overdiagnosis or medicalization are discussed, and the reliability of screening methods may be criticized, news reports are generally positive about the effectiveness of CSPs. Other issues regarding the organization of screening, and the psychological and physical impact received more negative attention. Examples are listed in Table 7.

## Discussion

As participation numbers in CSPs show a declining trend and research shows that news media portrayals might play a role in this, we investigated how news media in the Netherlands report on cancer screening. While we mainly focused on the country's three CSPs (i.e., breast, colorectal, and cervical), we did not exclude other types of cancer. We analyzed these news reports through a systematic content analysis. Our study had four main findings. First, we found that most news reports frame CSPs as effective, and that one in five reports dispute the effectiveness to some extent. This is in line with earlier work from other countries showing that news media encourage screening [60] and discuss potential expansions in the access to screening but simultaneously may undermine confidence in existing screening programs [29], or discuss considerations to a lesser extent [49]. Moreover, we found that most reports discuss breast cancer, which is in line with other work that finds that breast cancer is often overrepresented in news [29,61]. However, we found that the ratio of

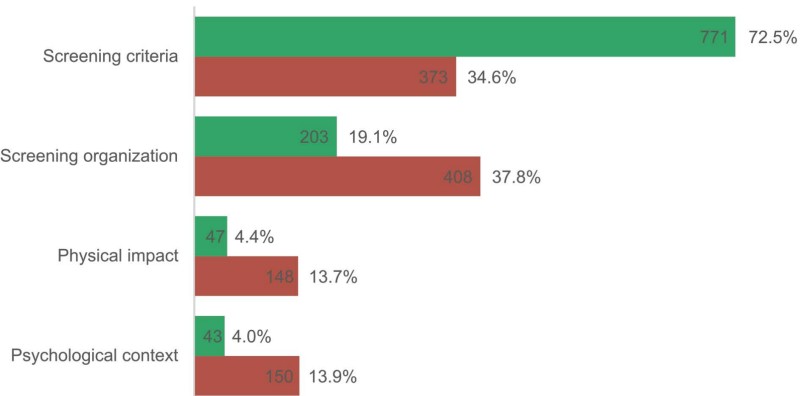

**Fig 4. Distribution (n and %) of the different argument types in favor or against cancer screening.** Percentages display the distribution of an argument type within all pros (n = 1,064) or cons (n = 1,079).

**Table 7. Examples of pros and cons in news media for cancer screening.**

| Argument type | Pros | Cons |
|---|---|---|
| Screening criteria | • "Women who participate in the national screening program have higher chances of successful treatment."<br>• "Handing in a small sample of your stool allows the early detection of colon cancer."<br>• "Screening can save many lives" | • "Many have to undergo screening to detect a relatively small sample of sick individuals."<br>• "Those with a negative test result do not have any guarantees of not having colon cancer. The current test is expected to miss one in three cases of colon cancer."<br>• "Screening turns healthy people into patients." |
| Screening organization | • "A home kit is available for those who do not want to 'take place in the chair.'"<br>• "Participation is for free and voluntarily."<br>• "Measures have been taken to screen safely within the COVID-19 safety protocols." | • "There is a shortage of physicians."<br>• "Screening is too expensive."<br>• "Screening had to be postponed due to COVID-19." |
| Physical impact | • "The radiation dose is considerably lower than it used to be."<br>• "The posture is inconvenient, but it does not bother me." | • "Some screening methods can be very invasive."<br>• "One in five women experience physical complaints after their pap smear."<br>• "An annoying viewing exercise via the anus, including a small chance on intestinal perforation." |
| Psychological context | • "Good news is always good, but with news that is less positive, people still experience relief as it provides them with clarity."<br>• "Every well-performed mammogram is of great value. It prevents a lot of stress and anxiety." | • "Someone's world is turned upside down when breast cancer is suspected, leading to an incredible amount of stress and fear."<br>• "It is hard to tell over a cup of coffee that your stool contains blood."<br>• "Women were afraid of the pap smear." |

NB: Cells with significantly higher relative frequency (pro vs. con or vice versa) are in bold.

reporting on a specific type of cancer and that specific CSP was highest for cervical cancer. This means that while most reports discuss breast cancer, reports discussing cervical cancer more often discussed the cervical CSP. These results show that the way that news media report on cancer screening in the Netherlands does not differ much compared to what is found in studies in other countries. Besides, we find that while breast cancer may be overrepresented in the news media, a relatively small portion of news reports discuss that cancer screening program. For cervical cancer, the portion of news reports on the topic that also discuss screening is higher, this might be seen as good news, as this is the screening program with the lowest participation numbers and where potential participants experience a lot of barriers [62]. News media may be able to inform the public to improve health-related decision making.

Second, even though we found no substantial peaks in the coverage of cancer screening over time, some specific minor news events existed. For instance, a peak in coverage was found around the time when advice for introducing a home kit for cervical cancer screening was announced. This compares with news media portrayals of other diseases like COVID-19 [63] or H1N1 [19], where peaks also existed around real-world events. This is in line with existing work on news values, stating that to become portrayed in the media, information needs to be considered newsworthy [64]. In addition, our results were also in contrast with other work showing that cancer screening search trends decreased during the COVID-19 pandemic [65]. Our work shows that the volume of news reports in our data around the outbreak of the pandemic remained stable. Oppositely, news reporting on breast cancer screening increased because the screening program was paused because of the pandemic. A broader reason for this contrasting finding might be that news consumption increased during the pandemic, but only for television, Internet, and social media [63]. Taken together, our results might imply that news companies might perceive stories on CSPs as newsworthy, but not as newsworthy as, for instance, the COVID-19 pandemic which led to increased news consumption. In other words, cancer screening seems newsworthy, but only to a certain degree, which makes that there is a base level of reporting on CSPs, that increases if a real-world event happens.

Third, confirming earlier work [66], we found that differences exist in news media reports regarding breast, cervical, and colon cancer screening. Regarding breast cancer screening, for instance, news frequently discusses the physical impact, such as pain, while for cervical cancer, arguments often relate to psychological aspects, such as taboos and lack of motivation. Breast cancer is most reported on in our sample, which is in line with other work [29,42]. However, breast cancer was the fifth type of cancer regarding absolute and relative mortalities in the Netherlands, as 6.9% of all cases of breast cancer lead to morbidity [67]. Thus, our results show that, again, news media reports on cancer screening might not be different in the Netherlands, compared to other countries [42]. However, this study gives a more granular overview and shows that for different CSPs, different arguments are used. Moreover, the risks of getting cancer are also different, where the risk of getting colon cancer is framed the largest (please note this is a trend in our data, not a significant difference). Besides, the consequences of cancer are framed as neutral or severe, where only a few articles describe the consequences as not severe, and this is only for breast (1.8%) and colon (2.7%) cancer. This shows that while news reports are sometimes critical about screening programs, they mostly frame cancer as a serious topic, which could also be why it is considered newsworthy.

Finally, we found that different sources framed CSPs differently and that there were differences in when pros and cons were presented. We found that news reports citing individuals with personal involvement tended to frame CSPs as effective, whereas professional sources tended to have a more critical stance towards the programs' effectiveness. While one of the goals of the National Institute for Public Health and the Environment is to increase the participation in CSPs, we would advise the organization to, in collaboration with the newspaper publishers and journalists, involve health care providers and people with personal experiences in the articles. Moreover, the current websites of the institute mostly provide participation and invitation numbers, but the number of people that are cured (indirectly) because they took part in CSPs may be communicated more, so the effects can be made more tangible, and the benefits are clearer.

Moreover, we found that the reliability of screening methods is sometimes criticized, and in such cases, news reports discuss overdiagnosis or medicalization. However, in line with earlier research, news reports are generally positive

regarding the effectiveness of CSPs and their benefits to public health [49]. This confirms findings in other work underlining that CSPs are framed as important [29].

### Limitations and future work

The purpose of this study was to describe news reporting on cancer screening, and not to test effects of news reports. However, we would like to point out that other research might be able to investigate those relations through cross-sectional experimental or longitudinal designs. These studies could then, for instance, establish causal relations of news media exposure on public perceptions, health-related information gathering or even intentions to take part in screening.

In this study, we made use of duplicate detection via the Rouge-metric and removed duplicates from our sample. We did this because articles in the Netherlands often are double posted (e.g., in national paper Algemeen Dagblad and in regional paper De Gelderlander, both part of the portfolio of DPG Media). We did this because most people do not read those newspapers simultaneously. Besides, most of those articles are direct duplicates, and not another article on the same topic. However, we do recognize that the volume of articles on certain topics or types of cancer is larger when those articles are not removed. These topics could then come across as potentially more salient. Future work might be interested in a more absolute comparison between different types of cancer and cancer screening in the (Dutch) media landscape.

With this work, we aimed to describe and scrutinize how written news media reports in the Netherlands talk about cancer and cancer screening in the country. While news media still play an important in setting the public agenda, research shows that this agenda is often overlapping with social media discussions, also in the case of cancer-related health communication [22,23]. Moreover, it is important to recognize that parts of the public do use social media platforms as their prime source of news media, either directly or by proxy (i.e., following news production companies or outlets on Facebook or Twitter). Thus, next to news media reports, future research could aim to draw stronger comparisons between traditional news and social media occurrences of cancer and cancer screening and information gathering or even participation in cancer screening. Additionally, we would like to underline that within this focus on traditional news, we focused on the content of the news articles and do not have information about the impact that news articles have. While implications of media impact have been reported in other research [68,69], we build on that research and do not have new evidence regarding impact. Future research may be able to investigate the impact by for instance, looking at the frames that are used in articles on cancer screening and how individuals potentially adjust their attitudes because of this.

Finally, we would like to point out that the COVID-19 pandemic had effects on our dataset. First, the incidence of registered breast cancer decreased during the pandemic. The reason for this is that the personnel that worked for the screening programs was allocated to help with the pandemic (e.g., to put less pressure on all people working in the Dutch healthcare system) [70]. Second, while the pandemic was part of everyday life starting in 2020, we recognize that COVID-19 news could have expelled news on cancer and cancer screening. This could be true for news in general, but also news on health and health care specifically. Future research might be able to investigate this by comparing studies on news reports before, during but also after the pandemic, as our current dataset lacks the final tail of COVID-19.

### Conclusion

News media reports have the potential to contribute to how people think about CSPs, which, in turn, potentially predicts partaking. The current work presents a systematic analysis of different aspects of Dutch news articles on CSPs. In news media reporting, CSPs are usually described as a 'necessary evil' in the sense that taking part may be inconvenient and stressful for individuals. However, the early detection and diagnosis of cancer are generally perceived as important, and these benefits seem to outweigh the necessary evil in news media reporting.

## Supporting information

**S1 File.  News media outlets and search string.**
(DOCX)

**S2 File.  Rouge test scores for duplicate detection.**
(DOCX)

**S3 File.  Frequency distribution of the news framing for all sources.**
(DOCX)

**S4 File.  News events related to cancer screening.**
(DOCX)

## Acknowledgments

We thank Lisa Salm and Noa ter Braak for their assistance with the data coding.

## Author contributions

**Conceptualization:** Inge Stortenbeker, Hanneke Hendriks, Suzan Verberne.

**Data curation:** Inge Stortenbeker.

**Formal analysis:** Inge Stortenbeker.

**Funding acquisition:** Hanneke Hendriks, Suzan Verberne, Gert-Jan de Bruijn, Enny Das.

**Methodology:** Inge Stortenbeker.

**Project administration:** Martin-Pieter Jansen, Inge Stortenbeker.

**Supervision:** Martin-Pieter Jansen, Hanneke Hendriks, Suzan Verberne, Gert-Jan de Bruijn, Enny Das.

**Validation:** Hanneke Hendriks, Suzan Verberne, Gert-Jan de Bruijn, Enny Das.

**Writing – original draft:** Martin-Pieter Jansen.

**Writing – review & editing:** Martin-Pieter Jansen, Hanneke Hendriks, Suzan Verberne, Gert-Jan de Bruijn, Enny Das.

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
