## [Decision Letter · Decision Letter 0]

8 Aug 2025

PONE-D-24-50452Unpleasant But Effective: News Media Coverage of Cancer Screening in the Netherlands from 2010 to 2022PLOS ONE

Dear Dr. Jansen,

Thank you for submitting your manuscript to PLOS ONE. After careful consideration, we feel that it has merit but does not fully meet PLOS ONE’s publication criteria as it currently stands. Therefore, we invite you to submit a revised version of the manuscript that addresses the points raised during the review process.

We look forward to receiving your revised manuscript.

Kind regards,

Reza Rostamzadeh

Academic Editor

PLOS ONE

Journal Requirements:

[Disclosure statement: This research is part of the SENTENCES (Social mEdia aNalysis To promotE cancer Screening) project and is funded by ZonMw under grant project number 555004205. Project leader for this project is prof. dr. HHJ Das, who is also an author of this paper. https://projecten.zonmw.nl/nl/project/sentences-social-media-analysis-promote-cancer-screening].

[We thank Lisa Salm and Noa ter Braak for their assistance with the data coding.

Declaration of interest statement: the authors declare no potential conflicts of interest with respect to the research, authorship, and/or publication of this article.

Disclosure statement: This research is part of the SENTENCES (Social mEdia aNalysis To promotE cancer Screening) project and is funded by ZonMw under grant project number 555004205.

Data availability statement: raw files, study protocol, analysis plan, and codebook can be found on OSF: https://osf.io/z7y6u]

[Disclosure statement: This research is part of the SENTENCES (Social mEdia aNalysis To promotE cancer Screening) project and is funded by ZonMw under grant project number 555004205. Project leader for this project is prof. dr. HHJ Das, who is also an author of this paper. https://projecten.zonmw.nl/nl/project/sentences-social-media-analysis-promote-cancer-screening ]

5. Please include captions for your Supporting Information files at the end of your manuscript, and update any in-text citations to match accordingly. Please see our Supporting Information guidelines for more information: http://journals.plos.org/plosone/s/supporting-information .

Reviewers' comments:

Reviewer's Responses to Questions

**Comments to the Author**

1. Is the manuscript technically sound, and do the data support the conclusions?

Reviewer #1: Yes

Reviewer #2: Yes

Reviewer #3: Partly

Reviewer #4: Partly

Reviewer #5: Yes

Reviewer #6: Yes

2. Has the statistical analysis been performed appropriately and rigorously? 

Reviewer #1: N/A

Reviewer #2: Yes

Reviewer #3: Yes

Reviewer #4: N/A

Reviewer #5: Yes

Reviewer #6: Yes

3. Have the authors made all data underlying the findings in their manuscript fully available?

Reviewer #1: Yes

Reviewer #2: No

Reviewer #3: No

Reviewer #4: No

Reviewer #5: Yes

Reviewer #6: Yes

4. Is the manuscript presented in an intelligible fashion and written in standard English?

Reviewer #1: Yes

Reviewer #2: No

Reviewer #3: Yes

Reviewer #4: Yes

Reviewer #5: Yes

Reviewer #6: Yes

5. Review Comments to the Author

Reviewer #1: The author Martin-Pieter Jansen and their colleagues made an interesting research article entitled "Unpleasant But Effective: News Media Coverage of Cancer Screening in the Netherlands from 2010 to 2022".

Overall, the manuscript is written well and addresses the news media coverage of cancer screening in Netherlands.

In this study, various aspects of Dutch news articles concerning CSPs are systematically analyzed. As a result of their inconvenient and stressful nature, CSPs are often referred to in the media as a 'necessary evil'. Reports on cancer early detection and diagnosis are generally perceived to be more beneficial than the necessary evil.

I recommend the manuscript for acceptance after minor revisions.

Comments:

Table:1, The author should make the table with suitable references.

Fig.1, 2, 3 and 4, the authors should keep legible fonts and clear TIFF images for all the figures.

There are several typos throughout the manuscript; the authors should rectify accordingly.

Font sizes were not even throughout the manuscript; also, the alignment is missing.

Reviewer #2: WPLOS ONE

PONE-D-24-50452

Unpleasant But Effective: News Media Coverage of Cancer Screening in the Netherlands from 2010 to 2022

Article type: Research Article

The research article by Jansen et al., 2024 is a very interesting and systematic analysis of the effect of media coverage/framing on cancer screening programs(CSP) and rate of participation in CSPs. Authors have systematically performed content analysis of 5500 articles reported in media between 2010 to 2022. Results showed that most media reports showcased CSP as effective and beneficial for health. Study also revealed the negativity aspects like physiological and psychological in framing CSP as this could decline participation rate in CSPs. Content analysis of the media reports provide a conclusive awareness to public health that participation in CSPs is important for early detection and diagnosis of cancer.

Few queries to authors

1. Which is prevalent cancer type among Dutch population based on the cancer incidence data reported between 2010 to 2022.

2. It is very difficult to make out the data from figure 3. Top left graph, what are those two trend lines in all the articles type data.

3. This study provides data on Breast, colon, prostate, Lung and Cervical cancer. Does it mean Dutch media framed only these cancer types.

4. In addition to the COVID pandemic situation, can authors implicate other situations leading to decline in breast cancer incidence.

5. To make the media to cover framing categories that might result in explicitly effective, what type of scientific data and information be disclosed from CSPs National Institute for Public Health and the Environment

Figures: Figures are not presented with appropriate figure legends

Figures are not provided in appropriate resolution, Images appear fuzzy as we zoom with texts are hazy.

Reviewer #3: It is necessary for the author to refresh the current citations. The article's expectations are not met by this particular work. It is essential for the author to compile further data on the detrimental elements, especially those related to physiological issues, and to substantiate this with organizational evidence. Besides focusing on breast, colorectal, and cervical cancers, the editor could have explored additional forms of cancer.

Reviewer #4: 1. Title - the title does not reflect the article content which includes media coverage of cancer as well as cancer screening; in addition it is important to reflect the type of media, which is limited to newspapers

2. The impact of the media coverage has not been sufficiently considered.

3. Who reads newspapers in the Netherlands? It is recognised in many countries that increasing numbers of people get their news from social media sources and newspaper reading is decreasing. What is the age profile of newspaper readers? The authors need to seriously consider this and reflect the situation in the Netherlands.

4. Also the omission of social media in methods and discussion is unusual. Why not considerered in this analysis?

5. The article is long and some editing would be useful. Sentence structure editing is needed in places.

6. Figures do not appear to be correctlky numbered or titled at the end of the document

Reviewer #5: I would like to congratulate the authors on collecting this valuable comprehensive dataset. It would be highly practical for decision-makers. The overall structure of the manuscript was well-organized and clear. Please find my minor comments below for your kind attention:

Theory:

Line68: Please change CPSs to CSPs

Line68: Please change Focusses to Focuses

Volume, characteristics, and content:

Line 106-107: “we aim to describe the frames used in the if pros and cons of screening are presented, and which arguments are used for this”. The sentence is grammatically unclear and needs rephrasing.

Differences across time and cancer types

Line 131: RQ2: “To what how does the media coverage change over time?” The sentence is grammatically unclear and should be revised.

Search procedure:

As mentioned, non-Dutch newspapers were excluded. However, the manuscript does not clarify whether scientific English-language articles from Dutch newspapers were considered. If no such English-language content exists, please state this explicitly.

Additionally, if there was uncertainty about this, it may have been helpful to include English keywords (e.g., cancer, colorectal cancer, colposcopy) in the search strings as well.

Contextualization of news (RQ4):

Line 405: Based on the context and Table 6, the phrase “less likely neutral” may be incorrect. “More likely neutral” appears to align better with the data presented.

Discussion:

Lines 456 to 458: The comparison was insightful, However, the information would be more accessible if presented visually — e.g., as a diagram or chart.

Line 458: Consider changing “we find” to “we found” to maintain consistency in tense.

Lines 470 to 472: Please clarify the contrast mentioned in the sentence: “In addition, our results were also in contrast with other work showing that cancer screening search trends decreased during the COVID-19 pandemic.” Based on both your data and previous studies, it seems that cancer-related news coverage declined during the COVID-19 pandemic, regardless of the country. If this is the case, there may not be a real contrast. Please elaborate on how your findings differ.

Line 474: Consider replacing “All in all” with a more formal phrase.

Limitations and future work:

Line 516: The word “than” should probably be replaced with “then”.

Acknowledgements:

This section would benefit from clearer justification and improved wording to align stylistically with the rest of the manuscript.

Table 4:

The categories Preventive Measures and Sources appear inconsistent in terms of total values and percentages.

Additionally, Alternative screening (e.g., full-body scan), mentioned in Line 246, is missing from the table. Please review and adjust the totals and percentages accordingly.

Table 5:

Cancer consequences: Neutral: The total amount should be corrected to 300 (53.5)

Table 6:

Cancer risk: Please double-check and reconsider the calculation.

Table 7:

Screening criteria: Cons: The statement “Those with a positive test result do not have any guarantees of not having colon cancer. The current test is expected to miss one in three cases.” may be misleading. If the point is about false negatives or limited sensitivity, the phrase should reference negative test results instead.

Good Luck

Reviewer #6: I appreciate the opportunity to review this article. The authors have conducted a systematic study of various aspects of Dutch news reporting about cancer and cancer screening. The motivation for the study is clear, the methods are well-described, and the results are well-presented. I think this article will provide useful descriptive information and methods that other researchers can build on in future work. I have only minor comments, which are detailed below.

Introduction

1. Lines 87-96: I wonder if these differences in findings could be due to cultural differences. Do the authors have any thoughts on this?

2. Lines 105-108: Typos/awkward wording, suggest rewording as: “Taking the above into account, the first aim of this work is to describe cancer related news, based on its volume and characteristics. Moreover, we aim to describe the frames used in cancer related news based on the pros and cons of screening that are presented, and which arguments are used for pros and cons. Combined, this is formulated in our first research question:”

3. Sentence beginning on line 116: My understanding is that breast cancer is the most common type of cancer in women, so is it really overrepresented? I think this point would benefit from a bit more detail, e.g., “overrepresented compared to its relative frequency” if that is the case.

4. Line 120-121: “Over representations” should be one word for consistency (“overrepresentations”

5. Line 121-122: I think this sentence would make more sense if the clauses were reversed: “The public perceptions of cancer do not always represent reality” or something similar.

6. Last paragraph on page 6, wording suggestions to make slightly more formal, starting at line 123: “However, to the best of our knowledge, no work that investigates news media reporting on cancer in the Dutch media landscape has yet been published. Therefore, the second aim of this work is to describe the coverage of cancer screening in written news media over time from 2010 until 2022. The third aim is to investigate differences in reporting between the three types of cancer screening. These aims lead to the second and thirs research questions:”

7. Line 131: RQ2 should start with either “to what extent” or “how,” or there should be an “and” that is missing.

8. Line 135: Suggest removing “that are” in this line.

9. Line 144: “Cancer risks factors” should be “cancer risk factors”

10. Line 149: Suggest changing “that” to “this”

11. Line 151: Can you use a more specific word than “things”?

12. First full paragraph on page 8: Do you need “and pros and cons”? I thought this was how frames were being characterized.

13. I think more detail is needed in either the description or specific wording of RQ4. Can you tell us what you mean by “how information is contextualized”? Does this mean who is being quoted in the articles? What articles appear next to it? Needs a little more explanation/clarity.

Method

14. Search procedure, line 173: Suggest adding the month in 2010 when search started (January?) since you include the month in 2022.

15. Line 235: Should “news” be “news reports”?

16. I appreciate the thoroughness of this section.

Results

17. Line 318: Suggest rewording to “Though this proportion may seem high, only a few articles explicitly reject screening…”

18. Line 402: Should “neutral” be “as neutral” or “neutrally”?

19. Line 428: Suggest rephrasing, awkward to start the sentence with “Meaning that…”

Discussion

20. Line 457: I don’t think you need the parentheses around “much.”

21. Line 460-463: Suggest splitting into two sentences.

22. Line 462: I think the word “can” should be “may.”

23. Line 493: Typo, “frames” should be “framed.”

24. Line 507: The suggestion to include celebrities in CSPs seems slightly out of place. I suggest either removing this or digging into the idea a bit more.

25. Lines 509-11: Similarly, I would either remove the point about informed consent or go into more detail on it.

26. Line 516: Typo, “than” should be “then.”

Figures

27. Figure 2: Would it be possible to have the place the N for cancer consistently to the right of the shaded bar? Having two different places for this N is confusing.

6. PLOS authors have the option to publish the peer review history of their article (what does this mean? ). If published, this will include your full peer review and any attached files.

**Do you want your identity to be public for this peer review?** For information about this choice, including consent withdrawal, please see our Privacy Policy .

Reviewer #1: **Yes: ** Dr.D.S.Prabakaran

Reviewer #2: No

Reviewer #3: **Yes: ** Kannappan Mohanvel Sucharitha

Reviewer #4: No

Reviewer #5: **Yes: ** Mahsa Javadi

Reviewer #6: No

---

## [Editor Report · Decision Letter 1]

23 Sep 2025

Unpleasant But Effective: Newspaper Coverage of Cancer Screening and Cancer in the Netherlands from 2010 to 2022

PONE-D-24-50452R1

Dear Dr. Jansen,

We’re pleased to inform you that your manuscript has been judged scientifically suitable for publication and will be formally accepted for publication once it meets all outstanding technical requirements.

Kind regards,

Reza Rostamzadeh

Academic Editor

PLOS ONE
---

## [Editor Report · Acceptance letter]

PONE-D-24-50452R1

PLOS ONE

Dear Dr. Jansen,

I'm pleased to inform you that your manuscript has been deemed suitable for publication in PLOS ONE. Congratulations! Your manuscript is now being handed over to our production team.

Kind regards,

on behalf of

Dr. Reza Rostamzadeh

Academic Editor

PLOS ONE